# Bioactivity and Sensory Properties of Probiotic Yogurt Fortified with Apple Pomace Flour

**DOI:** 10.3390/foods9060763

**Published:** 2020-06-10

**Authors:** Marina Jovanović, Marija Petrović, Jelena Miočinović, Snežana Zlatanović, Jovanka Laličić Petronijević, Dragana Mitić-Ćulafić, Stanislava Gorjanović

**Affiliations:** 1Institute of General and Physical Chemistry, Studentski trg 12/V, 11158 Belgrade, Serbia; marijapetrovic52@gmail.com (M.P.); snezana.zlatanovic@gmail.com (S.Z.); 2Faculty of Biology, University of Belgrade, Studentski trg 16, 11158 Belgrade, Serbia; mdragana@bio.bg.ac.rs; 3Faculty of Agriculture, University of Belgrade, Nemanjina 6, 11080 Belgrade, Serbia; jmiocin@agrif.bg.ac.rs (J.M.); jovankal@agrif.bg.ac.rs (J.L.P.)

**Keywords:** apple pomace flour, probiotic yogurt, sensory properties, antioxidant activity, cytotoxicity

## Abstract

To meet the demand for new functional foods in line with the trend of sustainable development, a novel probiotic yogurt fortified with 1%, 3%, and 5% apple pomace flour (APF) added immediately after inoculation with *Lactobacillus acidophilus, Streptococcus thermophilus,* and *Bifidobacterium bifidum* was developed. Upon fermentation in the presence of APF, a number of probiotic strains remained within the required range, while the syneresis of enriched yogurts was reduced up to 1.8 times in comparison to the control. Supernatants (i.e., extracted whey) obtained from yogurts with 1%, 3%, and 5% APF respectively had 1.4-, 1.8-, and 2.3-fold higher total phenolic content (TPC) than the control, 3.3-, 4.7-, and 8.0-fold higher radical scavenging (DPPH), and 1.3-, 1.6-, and 1.7-fold higher reducing activity (FRAP). Also, probiotic yogurt supernatants (3% and 5%) inhibited colon cancer cells’ viability (HCT 116, 12% and 17%; SW-620, 13% and 19%, respectively). The highest firmness, cohesiveness, and viscosity index values, and the highest scores for color and taste, were obtained for yogurt with 3% APF, indicating that this is the optimal APF amount for the production of novel yogurt with functional properties.

## 1. Introduction

United Nations Sustainable Development Goals aimed at ensuring sustainable consumption and production patterns, as well as the achievement of good health and well-being, have directed the flow of current research toward the recovery of agri-food byproducts and the exploration of their potential as functional and nutritional ingredients. The first published proposal for apple pomace (AP) usage in human nutrition appeared 48 years ago [1,2]. The re-emergence of the idea of introducing carefully processed AP in commercial food products, as one of the main food research trends currently [3], has been based on multifarious findings that have highlighted the positive economic aspects and health benefits of AP implementation. 

Large quantities of AP, a byproduct of the juice and vinegar processing industry, are disposed of in landfill and incinerators, which, in addition to the financial burden of disposal, has a detrimental effect on the environment [4,5]. From another point of view, apples (*Malus* sp.) are recognized as a significant part of the diet in various cultures [5,6], and it has been confirmed that AP is also safe for human consumption [7]. Moreover, preclinical studies have reported that consuming AP can contribute to the prevention, reduction, and elimination of numerous pathologies by exhibiting a positive effect on antioxidant status, metabolic dysfunction, improved gastrointestinal health, and weight loss. Also, AP decreases the risk of colon cancer [7]. Furthermore, AP may serve as a carrier of phytochemicals and a source of non-viable nutrients that stimulate the numbers and/or activities of probiotic bacteria, causing the limitation of intestinal pathogen growth [8].

Thus, a growing body of evidence supports the view that AP presents an extraordinary resource that can be used as a health-promoting food component. Moreover, AP waste management could respond to a decades-long challenge. So far, AP has been used as an ingredient in cakes, cookies, biscuits, pie filling, jellies, snacks, wheat bread, noodles, sausages, and yogurt [4,9,10,11,12,13]. Yogurt is one of the most popular food products with functional properties, commercially used as a carrier for gut-friendly bacteria and bioactive food ingredients [14]. It has been called a “functional food” due to the beneficial action of probiotic bacteria that engage in combat with pathogens, thus improving digestion and intestinal hygiene. To meet modern consumers’ needs, novel variants of yogurt-based products are constantly being investigated and introduced to the market, leading to a steady increase in the sales and popularity of this dairy product [15,16].

Studies on yogurt supplemented with minimally processed AP are scarce, and none of them refer to yogurt fortified with AP flour (APF) produced industrially by the innovative technological process developed recently (dehydration below 55 °C and grinding without heating) [17]. The thermal stability [18] and functional properties of APF [11], as well as the effects of its presence in a high fat and sucrose diet (HFSD) on glucose metabolism and obesity [19], were recently reported. Supplementation with APF (0.5% *w/w*) decreased body weight gain (BWG) and glycemia, and improved glucose tolerance, in mice exposed to an HFSD. A remarkable decrease in BWG was observed in mice exposed to the standard diet (SD) supplemented with APF. Food efficiency ratios calculated per BWG as well as per energy intake of HFSD and SD groups supplemented with only 0.5% *w/w* APF were highly reduced in comparison to HFSD and SD without APF, respectively. Thus, APF utilization in various formulations of food and dietary supplements has been suggested to be a reasonable approach in developing dietary strategies for the prevention of diet-driven diabetes and body weight management.

Up to the present, yogurts have been fortified with AP powder, fiber, and water extracts. Published research on AP-fortified yogurts has emphasized their adequate physico-chemical properties [4,15,20]. However, the biological activity and sensory properties of AP yogurts have been rather neglected. Although there are a few studies concerning bioactivity or sensory acceptance, they were conducted on yogurt fortified with AP extract [5], on fiber obtained from AP [21], or with a low concentration of AP powder, obtained by drying at laboratory-scale level [22]. 

As consumers’ focus is directed to both the sensory and health aspects of products, we aimed to methodically examine the bioactivity and sensory properties of a novel probiotic yogurt, fortified with different amounts of APF. In order to achieve this, a novel probiotic yogurt fortified with APF was designed and examined for physico-chemical, prebiotic, antimicrobial, and antioxidant (AO) properties, as well as the total content of phenolic compounds. The cytotoxic effects on selected human colon cancer cell lines were also surveyed. Moreover, changes in textural and sensorial properties related to APF proportion were monitored. Generally, the main goal of the research was to determine the optimal amount of APF to be added to yogurt, based on its functionality and sensorial properties. This study represents a step toward for work on fortifying fermented dairy products with the dietary fibers (DF), bioactive polyphenolics, and antioxidants present in added APF, without compromising the viability of used strains and the sensorial properties of the final product. 

## 2. Materials and Methods

### 2.1. Materials

AP, as material left over from juice production, containing mixed apple varieties, including Idared, Jonagold, Golden Delicious, and Granny Smith in a random ratio, was provided by the fruit processing company “Fruvita” (Smederevo, Serbia).

Pasteurized cow milk (dairy plant “Zapis Tare,” Serbia) was purchased from a local market (Belgrade, Serbia). 

The lyophilized starter culture FD DVS ABT-5, containing *Lactobacillus acidophilus*, *Streptococcus thermophilus* and *Bifidobacterium bifidum*, was obtained from Chr. Hansen (Hørsholm, Denmark). The bacterial cell cultures *Escherichia coli* (ATCC 25922), *Salmonella enterica* serovar Typhimurium (ATCC 14028), *Salmonella enterica* serovar Enteritidis (ATCC 13076), and *Listeria monocytogenes* (ATCC 19111) were obtained from the Department of Microbiology, Faculty of Biology, University of Belgrade, Serbia. 

The human colon cancer cell lines HCT 116 (ATCC CCL-247) and SW-620 (ATCC CCL-227) used in this study were kindly provided by Dr Tatjana Srdić-Rajić from the Institute of Oncology and Radiology, Belgrade, Serbia.

M17 agar, Bifidobacterium agar, De Man-Rogosa-Sharpe agar (MRS), and Müller Hinton agar (MHA) were obtained from HiMedia (Mumbai, India). Dulbecco’s modified Eagle’s medium (DMEM), penicillin–streptomycin mixtures, phosphate-buffered saline (PBS), trypsin from the porcine pancreas, 3-(4,5-dimethylthiazol-2-yl)-2,5-diphenyltetrazolium bromide (MTT), Trolox (6-hydroxy-2,5,7,8-tetramethylchromane-2-carboxylic acid), 2,4,6-tripyridyl-S-triazine (TPTZ), and gallic acid were purchased from Sigma-Aldrich (Steinheim, Germany). Folin–Ciocalteau reagent, sodium carbonate, sodium acetate trihydrate, acetic acid, hydrochloric acid, and sodium hydroxide were obtained from Merck (Darmstadt, Germany). DPPH (2,2-diphenyl-1-picrylhydrazyl) was produced by Fluka (Buchs, Switzerland). 

### 2.2. Production of APF

Apple pomace flour was prepared as described previously [17]. Sterilization of the APF was conducted with thermal ovens for research and industry, and the data were presented using AtmoCONTROL software (Appendix A). 

### 2.3. APF Water Extract Preparation 

APF in the amount of 5 g was submerged in water (100 mL) and incubated at 85 °C for 30 min. After overnight incubation with shaking at 37 °C, the sample was filtered, and the raw extract was evaporated to dryness in a vacuum evaporator (<45 °C) and dissolved in water to a final concentration of 100 mg/mL.

### 2.4. Production of Probiotic Yogurt Fortified with Apple Pomace Flours

Yogurts were prepared using pasteurized cow milk (Zapis Tare, Serbia) containing 2.8% of milk fat. The milk, heat-treated at 85 °C for 10 min, was cooled down to 43 °C, and a lyophilized starter culture ABF-5 (0.02%, 50 U, Chr. Hansen, Hørsholm, Denmark) containing *L. acidophilus*, *S. thermophilus*, and *B. bifidum* was added (initial counts of log 10.7 CFU/g for each of the containing bacteria strains). The inoculated milk was filled into glass containers (125 g). The selected concentrations of APF (1%, 3%, 5%, and 20% *w/w*) were added, and after thorough mixing, the samples were subjected to fermentation. Fermentation was set at 43 °C until pH 4.6 was reached (approximately 2.5 h). After fermentation, the yogurt samples were stirred and stabilized by cooling (4 °C for 24 h). Yogurts with 1%, 3%, and 5% APF were used in all further testing except in the cytotoxicity test. Yogurt containing 20% APF was further processed and used only in the cytotoxicity assay. All samples were made in triplicate and the samples analysis was repeated three times. 

#### 2.4.1. pH Measurement

pH was determined using a pH meter with a gel-filled electrode (WTW™ SenTix™ 41 pH, Massachusetts, MA, USA). 

#### 2.4.2. Syneresis of Yogurt 

Samples (2 × 25 g) from each batch were weighed in centrifuge tubes and centrifuged at 3000× *g* for 10 min at 4 °C. The whey was separated from the samples. The syneresis level was expressed as the weight of drained whey per 100 g yogurt.

### 2.5. Textural Properties

The textural properties of the yogurts such as firmness, cohesiveness, and index of viscosity were analyzed by a TA.XT Plus Texture analyzer (Stable Micro System, Godalming, Surrey, UK) through a single compression test, using a back extrusion cell disc (A/BE; diameter 35 mm; distance 30 mm; speed 0.001/ms) and an extension bar, using a 5  kg load cell at 5 °C. All measurements were performed at least in triplicate. 

### 2.6. Sensory Analysis

The overall sensory quality was determined using a scoring method (0–5). Yogurt samples were evaluated in terms of the most prominent sensory attributes, i.e., appearance (color), texture (creaminess and granulation), and flavor (odor and taste). For this purpose, a panel of nine members who completed the training for the selected assessor according to the requirements of the standard ISO 8586-1 were employed. The scores assigned by the panel have been weighted with corresponding coefficients of importance (CI) selected according to the influence of each attribute on overall sensory quality. By dividing the sum of weighted scores for each sample by the sum of the coefficients of importance (20), a quantitative quality indicator called the percentage of maximum overall quality was obtained. 

### 2.7. Preparation of Yogurt Supernatants 

Probiotic yogurt supernatants (i.e., extracted whey) were prepared as described previously [23]. Yogurt samples were centrifuged at 20,000× *g* for 60 min at 4 °C. The supernatants, filtered using a 0.45 µm syringe filter (MS® CA, Membrane Solution, Shanghai, China), were kept at −20 °C until further testing. Yogurt supernatant filtrates were obtained from a control yogurt (without APF) and yogurts with 1%, 3%, 5%, and 20% APF. The supernatant obtained from the control yogurt (without APF) and yogurts with 1%, 3%, and 5% were used for investigating total phenolic content (TPC), antioxidant assay, and disc diffusion assay. In the MTT assay, the supernatants obtained from the control yogurt and the yogurt with 20% APF were employed as initial stocks that were further diluted in DMEM to a desirable concentration.

### 2.8. The TPC and Antioxidant Assays

The procedures for the determination of TPC using the Folin–Ciocalteu (FC) method and AO activity using DPPH and ferric reducing antioxidant potential (FRAP) assays of prepared yogurt supernatants were described previously [24]. The results were expressed as g of gallic acid equivalents (GAE) per liter of supernatant for TPC, and mmol Trolox equivalents (TE) per liter of supernatant for AO capacity, measured by DPPH and FRAP assays.

### 2.9. Microbiological Analysis of Yogurts

#### 2.9.1. Bacterial Counts

*L. acidophilus*, *S. thermophilus,* and *B. bifidum* were counted using the pour plate technique and serial dilutions in phosphate-buffer saline (1% PBS). After anaerobic incubation at 37 °C for 72 h, plate counts of *B. bifidum* were performed in Bifidobacterium agar. *L. acidophilus* was counted on MRS agar (pH 6.2) containing 1 mg/L sorbitol, and the plates were anaerobically incubated at 37 °C for 72 h. *S. thermophilus* was enumerated using M17 agar (pH 7.2) under aerobic incubation at 37 °C for 48 h. Plates containing 30–200 colonies were counted, and the results were expressed as colony-forming units per mL (CFU/mL).

#### 2.9.2. Disc Diffusion Assay

To determine the antimicrobial activity in disc diffusion assay, the following species of bacteria were used: *E. coli* (ATCC 25922), *S.* Typhimurium (ATCC 14028), *S.* Enteritidis (ATCC 13076), and *L. monocytogenes* (ATCC 19111). The disc diffusion assay was carried out as previously reported [25]. Overnight bacterial cultures (100 µL) were spread onto MHA. APF yogurt supernatant filtrates (1%, 3%, and 5%) and APF water extract (WE, 50 mg/mL) were added into the well, and after incubation at 37 °C for 24 h, the diameter of the growth inhibition zones was measured. Streptomycin was used as a positive control.

### 2.10. Cell Culture Maintenance and Cytotoxicity Assay

The human colon cancer cell lines HCT 116 and SW-620 were maintained as a monolayer in DMEM supplemented with 10% fetal bovine serum, 1% penicillin/streptomycin mixtures, and 2 mM of L-glutamine.

The cytotoxic effect of supernatants of probiotic yogurts enriched with APF on HCT 116 and SW-620 were tested in parallel with control yogurts without APF and WE (initial stock 100 mg/mL), using an MTT assay as described [26]. The assay was performed after 24 h-long exposure of the cells to diluted supernatants of yogurt with 20% APF and control yogurt as well as diluted WE (50 mg/mL). The supernatant of yogurt with 20% APF was diluted in DMEM 20, 6.7, and 4 times in order to obtain concentrations equivalent to 1%, 3%, and 5% APF yogurt supernatants. The same dilutions of control yogurt supernatants, labeled as A, B and C, were prepared. 

### 2.11. Statistical Analysis

Data obtained from pH, syneresis, textural properties, sensory analysis, TPC and antioxidant assays, bacterial counts, and MTT assay were analyzed by analysis of variance (one-way ANOVA, Dunnett’s multiple comparisons test and Tukey’s multiple comparisons test) using GraphPad Prism software. The level of statistical significance was defined as *p* < 0.05.

## 3. Results and Discussion

### 3.1. Production of Novel Probiotic Yogurt Fortified with APF

From their first mention in 1972 [1,2] to the present, commercially available AP-based products have been limited in terms of both product diversity and manufacturer choice. In this context, products such as stirred-type probiotic yogurt fortified with APF produced by the recently developed technological process, dehydration below 55 °C and grinding without heating [17], have not been manufactured or even designed yet. The content of DF and total and individual phenolic compounds present in various APF samples including APF used within the scope of this study was determined previously. A high content of DF was ascribed to APF (45%), as well as a high TPC at 7.7 ± 0.3 mg, flavonoids at 24.8 ± 1.0 mg QE/g, and high AO activity [19].

In this research, different amounts (1%, 3%, 5%, and 20%) of sterilized APF were added and thoroughly mixed immediately after the inoculation of pasteurized cow milk with *L. acidophilus*, *S. thermophilus*, and *B. bifidum*. At the point when fermentation reached pH 4.6 (43 °C), the yogurt samples were stirred and stabilized by cooling (4 °C for 24 h). Furthermore, quality assessment and the biochemical and biological properties of novel probiotic yogurt fortified with APF were examined immediately after production. Yogurt containing 20% APF was further processed and used only in the cytotoxicity assay.

### 3.2. Quality Assessment of Novel Probiotic Yogurt Fortified with APF

#### 3.2.1. Influence of APF Addition on Yogurt pH and Syneresis

A statistically significant decrease in the initial pH (6.57) of pasteurized cow milk was detected in a dose-dependent manner after the addition of 1% (pH 6.28), 3% (pH 6.01), and 5% (pH 5.70) APF (Figure 1A). AP contains malic, ursolic, oleanolic, betulinic, chlorogenic, caffeic, p-coumaroylquinic, and ferulic acids [5,6,27]. Thus, the acidic character of APF due to the presence of natural acids could be responsible for this pH decrease [4]. Comparable to our results, do Espírito Santo et al. [22] and Fernandes et al. [5] reported the same response in pH change after the addition of 1% AP and 3.3% AP hot water extract, respectively. Furthermore, Staffolo et al. [15], do Espírito Santo et al. [22], Issar et al. [21], and Wang et al. [4] found that after a day of cold storage, the difference in pH value between control and AP-fortified yogurts was insignificant. This is also consistent with our findings.

Syneresis represents a significant defect of fermented dairy products such as yogurt and depends on numerous factors, including milk preparation, the coagulation process, and the ingredients used in production [28]. Data from the literature differ considerably on this issue. Staffolo et al. [15] reported that syneresis was absent in yogurt fortified with 1.3% apple fiber. Also, Wang et al. [4] argued that syneresis occurs only in yogurts supplemented with a lower percent of AP (0.5%). Different from the above but in accordance with [20], all the fortified yogurt samples in this study exhibited a reduction in their free whey portion in comparison with the control (Figure 1B). The inconsistencies in the literature data could be explained by differences in the AP used to enrich yogurt, yogurt types, and variations in the conditions applied during preparation. The gentle conditions applied in producing APF maintain a whole set of functional properties such as water and oil holding capacity, known to be related to the presence of soluble and insoluble fibers, respectively [11]. The total content of fiber in APF has been reported to be 45% [19] while pectine represented approximately 10% [18]. The high-methoxyl pectin present in APF could reduce syneresis and increase the viscosity of the continuous phase of the gel [4]. 

#### 3.2.2. Texture

An important criterion for the quality assessment of enriched yogurts is the texture, which depends on numerous factors, such as protein and fat content, production, added ingredients [29], and particularly, the dose and properties of fibers [15,30]. As expected, APF affected the textural properties of the yogurt significantly (Table 1). The highest firmness, cohesiveness, and viscosity index values of the yogurt with 3% APF suggest that this is the optimal dose for industrial production. The obtained results are in accordance with [20]. An improved texture was also reported for yogurts with incorporated byproducts such as passionfruit peel powder (0.5% and 1%) [31] and carrot cell wall particles (1% and 2%) [32]. Wang et al. [4] reported that an optimal concentration of AP is 0.5% for set-style yogurt preparation. The difference between the obtained and reported results could be explained by different yogurt types, preparation procedures, and properties of the ingredients added to enrich the product with DF.

#### 3.2.3. Sensory Evaluation

To the best of our knowledge, data on the sensory evaluation of yogurt fortified with minimally processed APF could not be found in the available literature. The only available information on the sensory properties of a product similar to ours indicated the acceptable sensory qualities of yogurt fortified with isolated apple fibers up to 5% [21]. In the present study, high scores for sensory properties were assigned to all analyzed samples by the panel. The highest rated was the sample with the addition of 3% APF (Table 2) because it gained similar ratings for creaminess, granulation, and odor as the control sample, as well as the highest scores for color and taste of all evaluated samples (Figure 2). It is worth noting that a portion of 100 mL of the yogurt with the highest score contains the optimal daily dose of APF, calculated using allometric scaling for dose conversion from mice to humans (approx. 3 g/d) [19]. Thus, regular, long-term consumption of that portion could be part of a dietary strategy for the prevention of diet-driven diabetes type 2 as well as obesity management.

### 3.3. Biochemical Properties: TPC and AO Activity

The TPC measured by Folin–Ciocalteu assay and the AO activity measured by DPPH and FRAP assay of the supernatants obtained from the probiotic yogurts with and without APF and WE are shown in Table 3. It can be observed that the addition of APF increases the TPC and AO activity of samples in a concentration-dependent manner. Corroborating the results presented here, a recent study demonstrated that an increase in AF content in yogurt samples is positively correlated with TPC [4]. Furthermore, in comparison to plain yogurt, yogurt supplemented with AP hot water extract exhibited significantly higher scavenging activity as measured by DPPH and ABTS assays [5]. According to the extensive literature data, AP alone demonstrates a strong AO potential. Wolfe et al. [33] showed that apple peel extract contained a high content of TPC and exhibited strong total AO activity. Also, an increase in AP extract concentration led to a significant elevation in AO activity [34].

Strong correlation coefficients between TPC and both DPPH and FRAP values (r = 0.98 and 0.87, respectively) were obtained, indicating that the polyphenols released from APF are the main contributors to the AO activity of the supernatants obtained from the APF-fortified yogurts. Phenolic acids (caffeic and chlorogenic acid), flavan-3-ols ((+)-catechin and (−)-epicatechin), flavonols (rutin), and dihydrochalcones (phloridzin) were found earlier in Granny Smith apple pomace [34,35], whereas a positive correlation was observed between the antiradical activities of pomaces from several apple varieties and the contents of total phenolic, flavanoid, flavan-3-ol, and some individual phenolic compounds [35]. The major phenols in APF (phlorizin, chlorogenic acid, quercetin) identified by high performance liquid chromatography coupled to diode array and mass spectrometer detectors (HPLC-DAD-MS/MS) were indicated to be the main contributors of prominent AO activity and antidiabetic and antiobesity agents that might be related to the metabolic effects of APF supplementation [19]. Phlorizin and chlorogenic acid content in various APF samples correlated with values obtained with DPPH, ABTS, and HPMC (Hydroxo Perhydroxo Mercury (II) Complex) antioxidant assays [19], which indicates that the contribution of both phenolic compounds to the total AO activity is significant.

Considering the gap between in vitro and in vivo studies, mechanisms of absorption and bioavailability occurring after consumption of AO-rich foods and beverages are not completely understood, thus representing a challenging area of research. In accordance with the directives of the European Food Safety Authority (EFSA) (Regulation (EC) No. 1924/2006 and Regulation (EU) No. 1169/2011 of the European Parliament and the Council), additional in vivo studies should be carried out to demonstrate the AO potential of APF-fortified yogurt and highlight its beneficial effect on consumers’ health [5]. Based on these premises, at this stage, APF-fortified yogurt can only be labeled as a potential functional beverage with improved AO properties. 

### 3.4. Biological Properties 

#### 3.4.1. Bacterial Viable Counts

Several studies have addressed the prebiotic activity of AP. However, this research has focused on specific carbohydrates obtained from AP [36], or on AP implemented in yogurt in low concentration (1%) [22] or in the form of AP extract [5].

Under the conditions of this experiment, *L. acidophilus* viable counts remained unchanged after the addition of APF in different concentrations. Although present, variation in *S. thermophilus* viability wase insignificant. *B. bifidum* viability slightly decreased with APF fortification in the amount of 3% and 5%, but its number was still within the recommended range for probiotic cultures (>log 7 CFU/g) (Table 4) [37]. In line with our results, do Espírito Santo et al. [22] reported that the viable counts of *S. thermophilus* and *L. acidophilus* did not differ in control and 1% AP-fortified yogurt. On the other hand, Fernandes et al. [5] demonstrated that in comparison to plain yogurt, yogurt supplemented with AP hot water extract (3.3%) decreased *S. thermophilus* counts at the end of fermentation. Furthermore, it has been shown that pectin-derived poly- and oligosaccharides isolated from AP exhibited a prebiotic effect, improving the vitality and adhesion of different lactic acid bacteria, while simultaneously reducing the adherence of various enteric pathogens [8,38]. 

#### 3.4.2. Antimicrobial Activity

The antimicrobial activity of the supernatants obtained from probiotic yogurts without and with APF (1%, 3%, and 5%) were analyzed against the food spoilage bacteria *E. coli*, *S. typhimurium*, *S. enteritidis*, and *L. monocytogenes*. WE (50 mg/mL) was also used. None of the tested samples showed antimicrobial potential against the selected pathogens at the tested concentrations. In contrast, Vodnar et al. [39] showed that methanol/water AP extract possesses antimicrobial activity against various Gram-positive and Gram-negative bacteria, including *E. coli*, *S. typhimurium*, and *L. monocytogenes.* This discrepancy could be related to differences in the extraction process that can lead to inconsistent chemical composition [27] and consequently to altered bioactivity. 

#### 3.4.3. Cytotoxic Properties

The cytotoxic potential of the supernatants obtained from probiotic yogurts without and with 20% APF (diluted to final concentrations of 1%, 3%, and 5%) and WE (final concentration 50 mg/mL) were tested on the human colon cancer cell lines HCT 116 and SW-620. In HCT 116, supernatants, 3% and 5%, and WE, 50 mg/mL, reduce cell viability for 12%, 17%, and 42%, respectively (Figure 3A). In SW-620, a reduction of 8%, 13%, 19%, and 37% viability was accomplished with supernatants, 1%, 3%, and 5%, and WE, 50 mg/mL, respectively (Figure 3B). Similarly, it was previously shown that apple phenolics and their gut-fermented products decrease the survival of the human colon cell lines LT97, HT29, and Caco-2 in a time- and dose-dependent manner [8,40]. Also, some in vivo studies confirmed that AP and yogurt have a cancer-preventing effect and support intestinal functions [3,41]. The proapoptotic effect, induction of suicidal cell death, was reported in mice treated with carcinogens and fed with yogurt. Additionally, both AP and yogurt could reduce levels of β-glucuronidase, which could be linked with a lower rate of colon cancer formation [7,41].

## 4. Conclusions

A probiotic yogurt enriched with apple pomace flour (APF), produced at industrial-scale level by a gentle and economically feasible process proposed recently, was developed. Here, the ingredient rich in DF and polyphenolics, APF, was added immediately after inoculation, but before fermentation, with the chosen probiotic strains. Such a practice led to a reduction of syneresis in comparison to the control, while a number of the probiotic strains used remained within the recommended range for probiotic cultures. The biochemical, biological, sensorial, and textural characteristics determined for enriched probiotic yogurts or their supernatants were related to the amount of APF added in a dose-dependent manner. As expected, the highest TPC, the most prominent AO, and cytotoxic activity against colon cancer cell lines were ascribed to supernatant obtained from probiotic yogurt with 5% APF. However, investigation of the sensorial and textural properties of probiotic yogurts revealed that the addition of 3% APF represents the optimal amount. Beside a high content of phytochemicals (1.8 times higher TPC than control), prominent AO activity (1.6 and 4.7 times higher FRAP and DPPH), and cytotoxicity on colon cancer cell lines HCT 116 (12%) and SW-620 (13%), supernatant obtained from probiotic yogurt with the addition of 3% APF or yogurt itself showed the best textural and most desirable sensory properties, and proved to be crucial in formulating and accepting each new product. Considering the content of APF in yogurt with the best sensorial properties and texture, the regular consumption of a portion of 100 mL per day can be recommended for better body weight control and diabetes prevention. All the presented results justify the use of APF-fortified probiotic yogurts in human nutrition and significantly contribute to general knowledge about the implementation of APF in the production of dairy products, the shortcomings of which are highlighted in [42]. However, quality changes in APF-fortified probiotic yogurts during storage should be examined in order to unambiguously confirm the distinct values of this novel product.

## 5. Patents

Patent publication number WO 2020/027683, Method for producing gluten-free flour made of apple pomace; 

Patent application number 2020/0618, Process for the preparation of a liquid or dehydrated probiotic fermented milk beverage with the addition of apple and/or beetroot flours predicted for human and dog nutrition. National Patent Application. 2020, Republic of Serbia.

## Figures and Tables

**Figure 1 foods-09-00763-f001:**
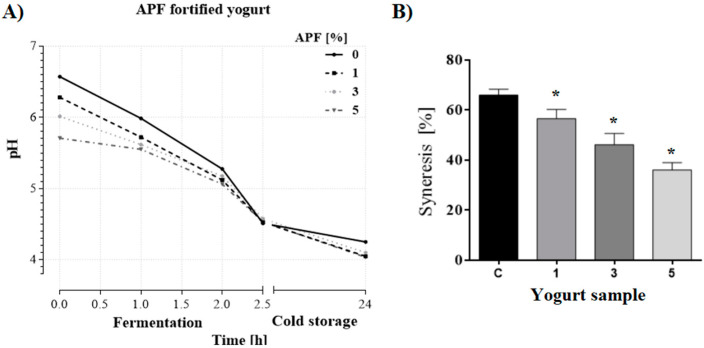
Decrease of pH value (**A**) and amount of whey segregated (%) from yogurt without (control—C; 0%) and with 1%, 3%, and 5% of apple pomace flour (APF) (1, 3, and 5) (**B**). Values are presented as mean ± standard deviation (* *p* < 0.05).

**Figure 2 foods-09-00763-f002:**
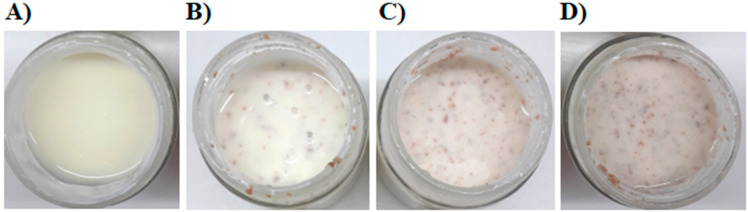
The yogurt samples without (**A**) and with 1% (**B**), 3% (**C**), and 5% (**D**) APF.

**Figure 3 foods-09-00763-f003:**
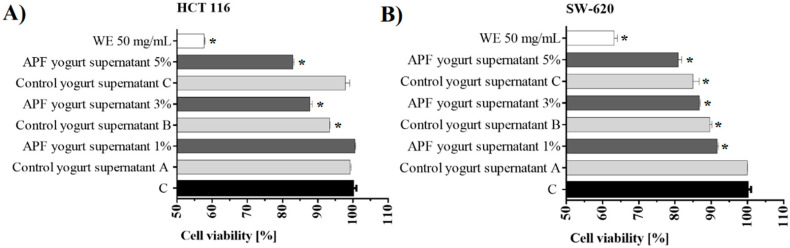
Inhibition rates of (**A**) HCT 116 and (**B**) SW-620 cells treated with supernatants obtained from probiotic yogurts with APF (APF yogurt supernatant 1%, 3%, and 5%) or without APF (control yogurt supernatants A, B, and C and WE (APF water extract)) after 24 h. C—control of cell viability (untreated cells) (* *p* < 0.05).

**Table 1 foods-09-00763-t001:** Textural parameters of APF-fortified yogurt.

Yogurt Sample ^1^	Firmness (g)	Cohesiveness (g)	Index of Viscosity (g s)
C	17.24 ± 1.49	11.4 ± 1.39 ^a^	4.08 ± 1.36 ^a^
1	16.14 ± 0.4	10.37 ± 0.05 ^b^	2.97 ± 0.04 ^b^
3	20.37 ± 1.47	14.15 ± 1.59 ^ab^	10.49 ± 4.71 ^ab^
5	19.85 ± 2.47	12.75 ± 0.93	6.15 ± 2.17

^1^ Control yogurt; 1, 3, and 5—yogurt made with 1%, 3%, and 5% apple pomace flour (APF); values in the table represent means of three replicated trials ± standard deviation. ^a,b^ Values with the same letter within the same column are significantly different in comparison to each other (*p* < 0.05).

**Table 2 foods-09-00763-t002:** Sensory evaluation of yogurts fortified with 1%, 3%, and 5% APF.

Sensory Attributes		Yogurt Sample ^1^			
	CI * (n)	C	1	3	5
Color	3	14.79 ± 0.19	14.14 ± 0.76	15.00 ± 0.00	14.57 ± 0.24
Creaminess	4	19.43 ± 0.24 ^ab^	14.29 ± 0.61 ^acd^	18.86 ± 0.27 ^ce^	17.43 ± 0.56 ^bde^
Granulation	3	13.50 ± 0.45 ^ab^	12.00 ± 0.27 ^a^	12.75 ± 0.46	11.50 ± 0.56 ^b^
Odor	3	14.33 ± 0.36 ^a^	12.67 ± 0.67 ^ab^	14.25 ± 0.43 ^b^	13.58 ± 0.36
Taste	7	31.46 ± 0.29 ^a^	28.39 ± 0.39 ^abc^	32.67 ± 0.35 ^b^	32.08 ± 0.47 ^c^
[%] of maximum overall quality	20	93.51	81.48	93.52	89.17

* CI—coefficient of importance; ^1^ Control yogurt; 1, 3, and 5—yogurt with 1%, 3%, and 5% of apple pomace flour (APF); values in the table represent means of three replicated trials ± standard deviation. ^a,b,c,d,e^ Values with the same letter within the same row are significantly different in comparison to each other (*p* < 0.05).

**Table 3 foods-09-00763-t003:** The total polyphenolic content (TPC), measured by the Folin–Ciocalteu (FC) method, and antioxidant (AO) activity of the supernatants obtained from probiotic yogurts with and without APF and water extract (WE) determined by 2,2-diphenyl-1-picrylhydrazyl (DPPH) and ferric reducing antioxidant potential (FRAP) tests.

Sample ^1^	TPC (mg GAE/L)	DPPH (mM TE)	FRAP (mM TE)
C	41.7 ± 0.3	0.03 ± 0.00	0.82 ± 0.02
1	56.3 ± 0.5 *	0.10 ± 0.00 *	1.10 ± 0.00 *
3	76.3 ± 1.7 *	0.14 ± 0.01 *	1.35 ± 0.01 *
5	96.3 ± 1.6 *	0.24 ± 0.00 *	1.38 ± 0.00 *
WE 50 mg/mL	206.4 ± 2.1 *	0.66 ± 0.00 *	1.88 ± 0.05 *

^1^ C—Control yogurt; 1, 3, and 5—yogurt made with 1%, 3%, and 5% apple pomace flour (APF); values are presented as mean ± standard deviation; asterisk (*) indicates significant differences between control and APF-supplemented yogurt samples (* *p* < 0.05).

**Table 4 foods-09-00763-t004:** Viable counts of *L. acidophilus*, *S. thermophilus* and *B. bifidum* in plain and APF-fortified yogurts.

Yogurt Sample ^1^	Viable Counts (log CFU/mL)		
	*L. acidophilus*	*S. thermophilus*	*B. bifidum*
C	8.67 ± 0.38	9.28 ± 0.79	9.14 ± 0.11
1	8.59 ± 0.27	9.03 ± 0.60	8.96 ± 0.24
3	8.65 ± 0.38	9.15 ± 0.48	8.76 ± 0.20 *
5	8.60 ± 0.27	9.18 ± 0.61	8.81 ± 0.15 *

^1^ C—Control yogurt; 1, 3, and 5—yogurt made with 1%, 3%, and 5% apple pomace flour (APF); asterisk (*) indicates significant differences between the control and APF-supplemented yogurt samples (* *p* < 0.05).

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
