# Peer review of "Bioactivity and Sensory Properties of Probiotic Yogurt Fortified with Apple Pomace Flour"

_foods, 2020, doi:10.3390/foods9060763_

Round 1

Reviewer 1 Report

This manuscript explore the sensory and bioactive effect of apple pomace flour (APF) supplementation to yogurt. The conclusion justify tha yogurt fortification wth APF. Although this paper is in general well done, I have some suggestions and requests:

1) minor spelling revisionsuch as at line 170 flavor instead of favor

2) line 144 please specify the % or the CFU inoculated of each strain

3) are the data reported in table 1 really statistically different? for exemple consistency has high standard deviations and they don't seem so different. please check

4) why there are 3 control in Fig. 2. Which is the diffrence between control a-b and c?

5) in the introduction the author sentence that 0.5% supplementation with APF has several beneficial implication. How the probiotic fortified yogurt fits this goal?

Reviewer 2 Report

The manuscript deals with development of a probiotic yogurt fortified with apple pomace; the samples were then investigated for their functional and sensory properties. In my opinion the study is interesting, even though not very innovative.

The two main concerns for recommending major revisions are the insufficient description of the method used for sensory analysis and the absence of results after a period of storage.

As to the first point, it is necessary to deepen the composition of the panel, training of the members and how the sensory attributes were chosen and "weighed".

As far as the second point is concerned, the authors should stress the fact that the whole study was done on fresh product (it appears that only one sampling was done, at 24 h refrigerated storage). So, they should clearly state that the discussion is done on quality of the "fresh product": it is well known, for instance, that cell viability must be measured at the end of shelf-life, the same from textural and sensory characteristics. Probably, the title itself should contain a word that clarify this aspect.

Reviewer 3 Report

The publication presents an innovative and interesting approach to the topic. The authors presented a lot of results, which in my opinion have a high scientific and application value.

Line 140: Please explain how many replications have been made of the analysis (n=?)?

Line 146: Please explain why 20% w/v APF was used? And how much was the dry substance of yogurt?

Line 298: Cohesiveness and Index of Viscosity values can be written as an absolute value.

Line 304: I suggest that the authors introduce an additional table with a description of selected descriptors.

Round 2

Reviewer 2 Report

The manuscript has been suitably modified